# Incidental Diagnosis of Urothelial Bladder Cancer: Associations with Overall Survival

**DOI:** 10.3390/cancers15030668

**Published:** 2023-01-21

**Authors:** Hubert Kamecki, Małgorzata Dębowska, Jan Poleszczuk, Tomasz Demkow, Artur Przewor, Łukasz Nyk, Roman Sosnowski

**Affiliations:** 1Department of Urogenital Cancer, Maria Skłodowska-Curie National Research Institute of Oncology, 02-781 Warsaw, Poland; 2Second Department of Urology, Centre of Postgraduate Medical Education, 01-809 Warsaw, Poland; 3Department of Computational Oncology, Maria Skłodowska-Curie National Research Institute of Oncology, 02-781 Warsaw, Poland; 4Nałęcz Institute of Biocybernetics and Biomedical Engineering, Polish Academy of Sciences, 02-109 Warsaw, Poland

**Keywords:** bladder cancer, screening, incidental diagnosis, survival

## Abstract

**Simple Summary:**

Bladder cancer prognosis is strictly related to the disease stage at diagnosis, suggesting that early detection could lead to improved treatment results. We retrospectively investigated for a possible association between incidental bladder tumor diagnosis and survival. We managed to demonstrate that patients who had been diagnosed incidentally, compared to non-incidentally diagnosed cases, tended to have improved survival, especially if the bladder lesion was first visualized with an ultrasound. However, although we did note marked survival benefit with incidental diagnosis in the subgroup of patients with a muscle-invasive disease, our results show that improved survival in the overall group of patients might have been caused by low-grade cancer overdiagnosis. This study serves as important evidence in the discussion on the possible role of screening for bladder cancer.

**Abstract:**

Background: We investigated whether an incidental diagnosis (ID) of bladder cancer (BC) was associated with improved survival. Methods: We retrospectively reviewed data of consecutive patients with no prior diagnosis of urothelial cancer who underwent a primary transurethral resection of bladder tumor (pTURBT) between January 2013 and February 2021 and were subsequently diagnosed with urothelial BC. The type of diagnosis (incidental or non-incidental) was identified. Overall, relative, recurrence-free, and progression-free survival rates (OS, RS, RFS, and PFS) after pTURBT were evaluated using the Kaplan–Meier curves and long-rank tests. A multivariable Cox regression model for the overall mortality was developed. Results: A total of 435 patients were enrolled. The median follow-up was 2.7 years. ID cases were more likely to be low-grade (LG) and non-muscle-invasive. ID vs. non-ID was associated with a trend toward an improved 7-year OS (66% vs. 49%, *p* = 0.092) and a significantly improved 7-year OS, if incidental cases were limited to ultrasound-detected tumors (75% vs. 49%, *p* = 0.013). ID was associated with improved survival among muscle-invasive BC (MIBC) patients (3-year RS: 97% vs. 23%, *p* < 0.001), but not among other subgroups stratified according to disease stage or grade. In multivariable analysis, only age, MIBC, and high-grade (HG) cancer demonstrated an association with mortality. PFS and RFS among non-MIBC patients did not differ in regard to the type of diagnosis. Conclusions: Incidental diagnosis may contribute to an improved survival in BC patients, most probably in the mechanism of the relative downgrading of the disease, including the possible overdiagnosis of LG tumors. Nevertheless, in the subgroup analyses, we noted marked survival benefits in MIBC cases. Further prospective studies are warranted to gain a deeper understanding of the observed associations.

## 1. Introduction

It is estimated that bladder cancer (BC) leads to more than 200,000 annual deaths worldwide [1]. While the prognosis in non-muscle-invasive BC (NMIBC) may be considered relatively favorable, especially in lower risk subgroups [2], only 36% of muscle-invasive BC (MIBC) patients were reported to survive 10 years after radical cystectomy (RC) with neoadjuvant chemotherapy [3], and the estimated 5-year overall survival (OS) in metastatic BC is as low as 5% [4]. This evident stage-dependent character of prognosis in BC patients leads to a hypothesis that an early diagnosis and prompt treatment, ideally prior to muscle invasion, could represent a strategy to reduce the large mortality burden of the disease. However, despite the fact that several BC screening trials have been reported in recent decades [5], high-level evidence in support of an association between screening or early detection of BC and survival is missing in the literature.

In our previous study we hypothesized that the natural history of BC may consist of an asymptomatic phase, representing an earlier stage of the disease, and we managed to demonstrate that incidental diagnosis of BC was associated with significantly reduced odds of a more advanced disease, regardless of the carcinoma grade [6], which we considered suggestive of a possible impact on prognosis. Now, the aim of this study is to collect the survival data of BC patients and to evaluate for a possible association between incidental diagnosis and prognosis.

## 2. Materials and Methods

A retrospective analysis of consecutive patients with no prior diagnosis of urothelial carcinoma, who underwent transurethral resection of bladder tumor (TURBT) at Maria Skłodowska-Curie National Research Institute of Oncology (MSCNRIO) in Warsaw (Poland), between January 2013 and February 2021, was performed. Inclusion criteria were: (i) having been diagnosed with urothelial BC, based on the TURBT pathology report and (ii) having a known type of diagnosis (incidental or non-incidental). Data in regard to demographics (birth date and sex), past medical history (cause of tumor diagnosis and tool used for diagnosis), diagnosis (pathology reports and imaging studies reports), and survival were collected. Specimens and imaging studies were not reviewed for the purpose of this study. Patients with missing or incomplete data were excluded from the analysis.

### 2.1. Incidental Diagnosis

We defined incidental diagnosis as a bladder lesion having been detected at a diagnostic study or examination performed for a reason other than evaluating BC-suggesting symptoms (gross or microscopic hematuria, non-infectious irritative voiding, urinary retention, renal colic, kidney failure, pelvic pain, anemia, or unintentional body weight loss).

### 2.2. Primary Disease Stage and Grade

Disease grade and local stage was determined based on: (i) primary TURBT (pTURBT) pathology report, (ii) second resection pathology report, if performed within 90 days after pTURBT and led to upgrading or upstaging, or (iii) cystectomy pathology report, if performed within 90 days after pTURBT and led to upgrading or upstaging. The primary stage was recognized as Tx if: (i) the patient was diagnosed with T1 cancer and no detrusor muscle was present in the specimen, as well as in any other specimen collected within 90 days after the T1 diagnosis; (ii) if pTURBT was incomplete and the tumor had not been completely excised within the next 90 days; or (iii) if the pathology report was equivocal. We considered the primary stage as metastatic in cases of either nodal or distant metastases.

All pathology examinations were performed by MSCNRIO pathologist dedicated to urogenital cancer and were reported in line with the International Society of Urological Pathology (ISUP) guidelines. Nodal and distant staging was determined based on imaging studies. For this purpose, we used reports of scans performed at MSCNRIO, if available; copies of external studies reports were used in other cases.

### 2.3. Other Definitions

Recurrence was defined as detection of intravesical lesion that was later pathologically confirmed to be recurrence of urothelial cancer, with lesions detected fewer than 90 days after pTURBT being considered incomplete resection, not recurrence. Progression was defined as either pathologically confirmed HG cancer in a patient previously diagnosed with LG cancer, or diagnosis of MIBC in a patient previously diagnosed with NMIBC, or diagnosis of metastases (either nodal or visceral) in a previously non-metastatic patient.

### 2.4. Outcome Measurements and Statistical Analysis

Data in regard to patient and disease characteristics were expressed as medians (with interquartile ranges [IQRs]) or numbers (with percentages). Mann–Whitney and χ^2^ or Fisher exact tests were used to compare continuous variables and percentages, respectively.

Overall survival (OS) and relative survival (RS) from pTURBT was evaluated. RS was calculated using the Pohar-Perme estimator [7], using the Polish annual life tables provided by Statistics Poland. Additionally, 3-year recurrence and progression free survival rates (RFS and PFS) were calculated for NMIBC patients. Survival analyses, stratified by type of diagnosis or disease characteristics, were performed using Kaplan–Meier curves with log-rank test, which was used for statistical comparisons. Survival rates were reported as percentages. A multivariable Cox regression model for overall mortality was developed and the results were expressed as hazard ratios with 95% confidence intervals. 

All statistical analyses were computed using R (version 4.2.0) and Matlab (MathWorks, version 2021a). Results were considered statistically significant if *p* < 0.05.

## 3. Results

Out of 744 patients, 435 met the inclusion criteria and were enrolled in the analysis (for the flowchart, see Appendix A). The comparison of included patients and patients excluded due to an unknown diagnosis type (*n* = 63) did not differ in terms of the primary disease stage; however, HG cancer was more prevalent in the included group (Appendix A). The patient and disease characteristics of the included patients, stratified according to the type of diagnosis (incidental or non-incidental), are provided and compared in Table 1.

In the non-incidental group, a bladder tumor was most commonly detected as a result of a diagnostic evaluation for gross hematuria (91%, Appendix A). Among the incidental diagnosis patients, in 83 (66%) the tumor had been initially detected with an ultrasound. Univariable comparison of ultrasound vs. non-ultrasound incidental diagnosis cases in regard to the primary grade or stage did not result in statistically significant differences (Appendix A).

The median survival follow up after pTURBT was 2.7 years (IQR: 1.0–4.7). In the 7-year OS and RS analysis (Figure 1A,B) there was a trend toward improved survival with incidental diagnosis, although the differences were non-significant; with 7-year OS and RS rates for incidental vs. non-incidental diagnosis being 66% vs. 49% (*p* = 0.092) and 84% vs. 65% (*p* = 0.263), respectively. However, when we limited the survival data to a 2-year follow-up (Figure 1C,D), the differences reached statistical significance, with 2-year OS and RS rates for incidental vs. non-incidental diagnosis being 84% vs. 73% (*p* = 0.039) and 91% vs. 80% (*p* = 0.023), respectively.

Tailoring the incidental diagnosis group to patients for whom the tumor had been initially detected with ultrasound improved the survival rates, with 7-year OS and RS being significantly superior compared to non-incidental diagnosis patients (Figure 2). The 7-year OS and RS rates for incidental with ultrasound vs. non-incidental diagnosis were 75% vs. 49% (*p* = 0.013) and 96% vs. 65% (*p* = 0.025), respectively.

The survival analyses stratified by disease characteristics demonstrated both the 7-year OS and RS to be significantly superior in patients with LG vs. HG cancer, as well as in cases of non-metastatic HG NMIBC vs. non-metastatic MIBC (Figure 3, *p* < 0.001 in all analyses).

Survival analysis of HG patients stratified by the type of diagnosis revealed no differences in the 7-year OS and RS between incidental and non-incidental diagnosis subgroups (Figure 4). The difference remained non-significant even after the incidental diagnosis subgroup was tailored to patients with ultrasound-detected tumors only (Appendix A).

The 7-year OS and RS analyses for non-metastatic HG NMIBC patients stratified by the type of diagnosis demonstrated a modest, non-significant trend toward inferior survival with incidental diagnosis (*p* = 0.172 for OS and *p* = 0.228 for RS, Figure 5A,B). The analyses performed for non-metastatic MIBC patients, with the follow-up limited to 3 years in view of a small numbers of patients at risk, revealed that the survival of incidentally diagnosed cases may have been superior (Figure 5C,D), with 3-year OS and RS for incidental vs. non-incidental diagnosis being 71% vs. 21% (*p* = 0.053) and 97% vs. 23% (*p* < 0.001), respectively.

Among the NMIBC patients, the 3-year RFS and PFS for incidental vs. non-incidental diagnosis were 56% vs. 52% and 91% vs. 87%, respectively; however, the differences were not statistically significant (Figure 6).

On multivariable Cox regression analysis that included age, sex, cancer grade (HG or LG), local stage (MIBC or NMIBC), distant stage (metastatic or non-metastatic), and incidental diagnosis, the factors which demonstrated a statistically significant association with overall mortality during the follow-up were older age, MIBC, and HG. (Table 2).

## 4. Discussion

This is the first study to report associations between incidental diagnosis and survival in urothelial bladder cancer patients. We demonstrated that while any observed beneficial impact of incidental diagnosis on survival cannot be extrapolated to the overall BC patient group in the long-term follow up, distinct subgroups of patients may be identified, for whom the association between incidental diagnosis and survival would be of greater significance.

In general, in spite of a suggestive trend that could have been observed on a curve analyses, the 7-year survival was not significantly improved with incidental diagnosis in the overall patient group. Although using a follow-up length cutoff of 2 years did result in reaching statistical significance for the observed difference, the clinical significance of this sole finding is low. Interestingly, the 7-year survival in the incidental diagnosis group became markedly and significantly improved only if the tumors detected with an ultrasound were included in the analysis. This result is difficult to elucidate, especially considering the fact that ultrasound and non-ultrasound incidental diagnosis cases did not significantly differ in regard to the primary stage or grade of the disease. A possible explanation is that non-ultrasound cases, comprised predominantly of patients for whom the tumor had been diagnosed with computed tomography (CT), might have had significant comorbidities, especially another cancer. This is likely, in view of the character of MSCNRIO as an institution and our department serving as the primary reference center for other-cancer patients diagnosed with a bladder tumor, as such a tumor could have been detected at a CT scan performed as part of follow-up for another malignancy. This issue could be recognized as a bias or a limitation; however, it may also allow us to consider the ultrasound cases to be more approximate to the “true”, real-life incidental BC diagnosis. Another explanation of the observed difference would be that a CT scan performed for non-urologic indication does not consist of a urography phase on a routine basis, and a non-urography CT scan is not recognized as a tool for bladder imaging [8], as opposed to the solid diagnostic performance achieved with a bladder ultrasound [9]. In light of the above discussion, our results may appear to be of marked significance if the bladder ultrasound, a relatively accessible diagnostic modality, is ever considered for BC screening purposes.

As expected, in our patients, a diagnosis of LG cancer in comparison to HG cancer and a diagnosis of HG NMIBC as compared to MIBC were associated with markedly improved survival. Given that those good prognostic features were significantly more frequent in the incidental diagnosis group, this could serve as a first-line explanation for the observed survival differences in the overall patient group, especially considering the fact that HG cancer and MIBC were demonstrated to be associated with overall mortality on the Cox regression model and incidental diagnosis was not. In general, most probably it was the relative downgrading and downstaging with incidental diagnosis and not incidental diagnosis itself that was responsible for most of the observed effects. We do not assume that having been incidentally diagnosed influenced the treatment decisions.

Interestingly, in the analysis tailored to HG patients, even in view of MIBC being significantly less common with incidental diagnosis in this subgroup, a similar survival was demonstrated between incidental and non-incidental diagnosis patients. Further analysis revealed two possible counteracting associations. The interesting trend toward inferior survival with incidental diagnosis in HG NMIBC is difficult to explain, and whether an association between a more malignant character of a lesion and its smaller tendency to bleed existed, or whether an incidental diagnosis was in fact a late diagnosis due to a prolonged lack of symptoms in those patients, remains subject to speculation. Importantly, the significantly improved survival with incidental diagnosis among MIBC patients appears to be much more relevant from the clinical point of view. However, these results should be interpreted with caution, as the analysis involved only up to 3 years of observation, and the sample size was relatively small, reflecting the rarity of incidental MIBC diagnosis.

Despite the demonstrated beneficial effects of an incidental diagnosis on 3-year OS and RS in MIBC, the above considerations lead to the conclusion that the major factors contributing to possibly improved survival in the overall patient group were the significantly higher rates of LG cancer in the incidental diagnosis subgroup, or the relative downgrading of the disease. In our previous study, we demonstrated that among NMIBC patients, incidental diagnosis was independently associated with LG disease [6]. As a small amount of untreated LG tumors possess the ability to progress [2], this could lead to a hypothesis that early diagnosis may have impacted survival in original LG patients for whom the natural history of the disease involved the risk of progression. Nevertheless, due to most probably a very small size of the effect, we consider this mechanism unlikely to be responsible for the observed results. Moreover, given that LG cancer is relatively indolent, and some authors consider it eligible for active surveillance [10,11], important concerns arise, whether incidental diagnosis might in fact have been responsible for the unnecessary detection of tumors with small malignant potentials, similar to prostate-specific antigen screening having led to an overdiagnosis of low-risk prostate cancer cases [12].

Our study provides important evidence for the discussion on the rationale of possible bladder cancer screening. Over the past decades, several tools have been proposed for this purpose, including urine dipstick or urinalysis alone [13,14], urine cytology alone [15,16], cystoscopy [17], biomarkers alone [18], or biomarkers combined with other non-imaging tests [19,20,21,22,23]. As we provide data in regard to a relatively large group of patients for whom an incidental diagnosis of bladder tumor was performed with an ultrasound, our results may be interpreted in the context of possible ultrasound-based screening strategies. Given that the resolution of an ultrasound is limited [24], and in the view that abdominal and pelvic imaging requires the use of a low-frequency transducer, most probably an ultrasound screening program would not result in a high yield of very small or in situ tumors. The addition of contrast-enhanced imaging to the protocol might result in slightly improved detection rates [25]. However, in our study, it was the advanced disease, namely MIBC, in which we noted the most marked survival benefits with incidental diagnosis. Further research is needed in order to gain a deeper understanding of this problem.

Interestingly, both 3-year RFS and PFS among NMIBC patients did not differ when stratified by the type of diagnosis. A possible explanation is that while incidental diagnosis was associated with relative disease downgrading and downstaging, the type of diagnosis did not impact treatment decisions in same-stage patients, and thus the further course of the disease was similar. 

In our study, we lacked data in regard to cancer-specific survival, which can be considered a major limitation. As mentioned above, many of our patients harbored another malignant disease; therefore, cancer-specific survival analysis would yield a clearer insight into the association between incidental diagnosis and prognosis. Assuming that an imaging study, which led to an incidental detection of a bladder tumor, had been likely performed for another, even non-malignant medical condition, this should raise attention to the fact that incidental diagnosis patients might have initially been at an increased risk of death due to possible comorbidities. Moreover, comorbidities, especially having been diagnosed with other malignancies, might have influenced BC treatment decisions, possibly leading to an increased reluctance toward more aggressive or radical treatment, and we lack the reliable data to perform a statistical analysis in regard to this issue. However, if true, the above-discussed hypothetical associations would have diminished rather than exaggerated the observed differences in survival. Nevertheless, significant mortality observed in HG and MIBC patients, as opposed to LG and NMIBC cases, suggests that bladder cancer was the major cause of mortality in our patients.

In order to partially overcome the above-discussed limitation, at least in regard to the possible impact of age or sex on overall survival, we performed the relative survival analyses, which allow for an estimation of survival independent of the national general population mortality [7]. We consider this a strength of our study.

The high rate of incidental diagnoses in our patient group, especially among HG tumors, may seem intriguing. However, as demonstrated in other reports from Poland [26,27], as well as in a large British study [28], the number of BC patients presenting without hematuria or diagnosed incidentally may indeed be higher than suggested by common clinical sense. Importantly, as symptomatic patients still may not undergo further diagnostic evaluation for various reasons [27], in order to best resemble screening-detected BC cases, we defined incidental diagnosis as tumor detection at a study performed for a reason other than aiming to explain possible BC symptoms. In line with this definition, incidental diagnosis might have also occurred in a patient who did experience symptoms, but those symptoms were not further evaluated, and the incidental detection was the earliest opportunity for establishing the diagnosis.

Apart from the above considerations, our results are prone to several other limitations, mainly resulting from the retrospective design of the study. Most importantly, a large proportion of patients were not enrolled in the analyses, due to the lack of data in regard to the type of diagnosis (incidental or non-incidental). While the comparison of the included and excluded patients did not result in finding significant differences in regard to the primary disease stage, the excluded patients demonstrated higher rates of LG cancer, and so we should assume that a selection bias might have occurred. Observer bias resulting in incorrect retrospective identification of diagnosis type also could have influenced our results. We lacked data in regard to tumor characteristics, e.g., tumor size and number, which would be helpful for the further stratification of subgroups. Relatively short observation times, as well as small sample sizes in the subgroup analyses, amount to another significant limitation.

## 5. Conclusions

This is the first study in regard to the possible impact of incidental tumor diagnosis on the survival of urothelial bladder cancer patients. We present evidence demonstrating that incidental diagnosis may be associated with improved overall and relative survival, especially in the cases of tumors detected with an ultrasound; however, most of the observed effects are probably a result of LG cancer overdiagnosis. Incidental diagnosis, while associated with relative disease downstaging and downgrading, had no independent impact on survival. Nevertheless, our results suggest that specific subgroups of patients may exist, e.g., MIBC patients, for whom incidental diagnosis may be of benefit. Further, preferably prospective studies are warranted before any conclusions in regard to possible BC screening strategies can be made.

## Figures and Tables

**Figure 1 cancers-15-00668-f001:**
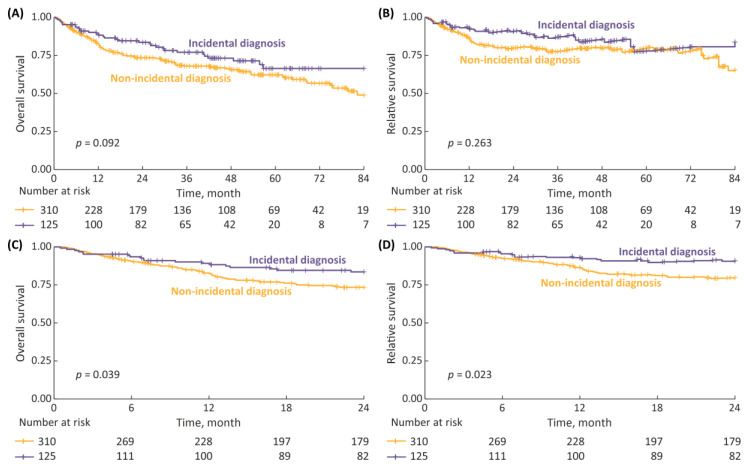
(**A**) 7-year overall survival (OS), stratified according to type of diagnosis (incidental vs. non-incidental); (**B**) 7-year relative survival (RS); (**C**) 2-year OS; and (**D**) 2-year RS.

**Figure 2 cancers-15-00668-f002:**
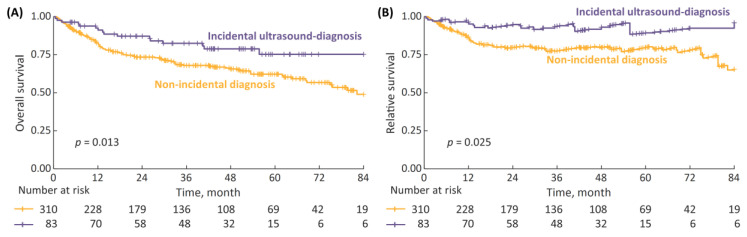
(**A**) 7-year overall survival (OS), stratified according to type of diagnosis: incidental diagnosis performed with ultrasound vs. non-incidental diagnosis; and (**B**) 7-year relative survival (RS).

**Figure 3 cancers-15-00668-f003:**
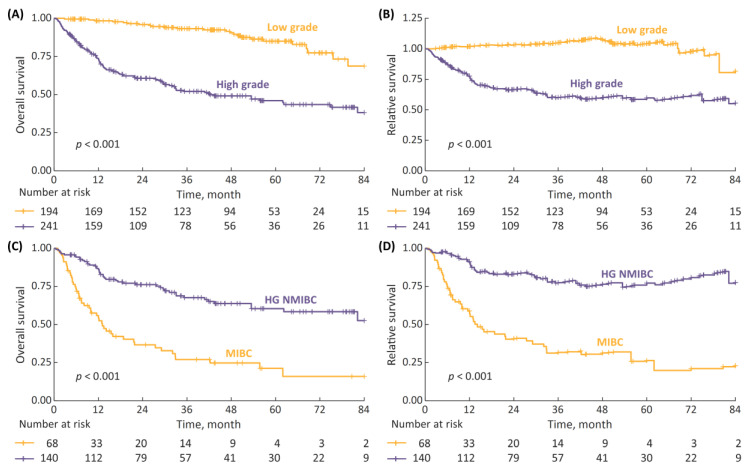
(**A**) 7-year overall survival (OS) stratified according to cancer grade (low-grade (LG) vs. high-grade (HG)); (**B**) 7-year relative survival (RS) stratified according to cancer grade; (**C**) 7-year OS in non-metastatic HG patients stratified according to disease stage (non-muscle-invasive bladder cancer (NMIBC) vs. muscle-invasive-bladder cancer (MIBC)); and (**D**) 7-year RS according to disease stage.

**Figure 4 cancers-15-00668-f004:**
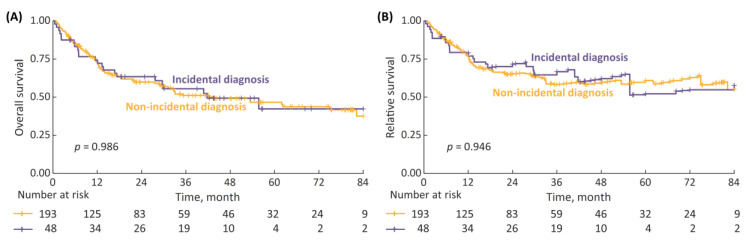
(**A**) 7-year overall survival (OS) in high-grade (HG) patients stratified by type of diagnosis (incidental vs. non-incidental); and (**B**) 7-year RS.

**Figure 5 cancers-15-00668-f005:**
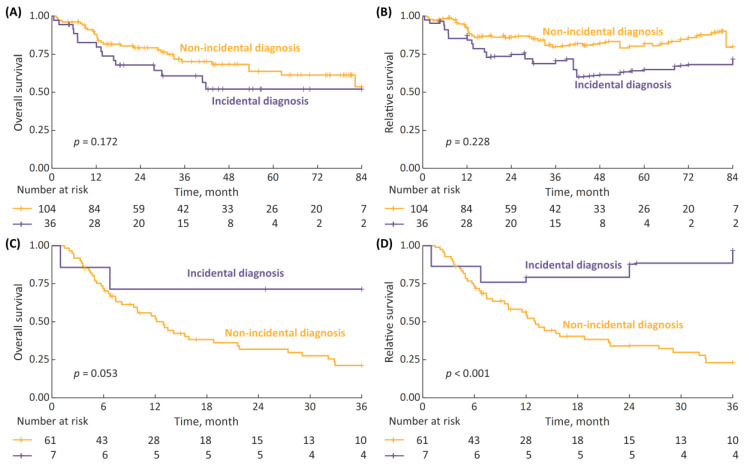
(**A**) 7-year overall survival (OS) in non-metastatic high-grade (HG) non-muscle-invasive bladder cancer (NMIBC) patients stratified by type of diagnosis (incidental vs. non-incidental); (**B**) 7-year relative survival (RS); (**C**) 3-year OS in non-metastatic MIBC patients stratified by type of diagnosis; and (**D**) 7-year RS.

**Figure 6 cancers-15-00668-f006:**
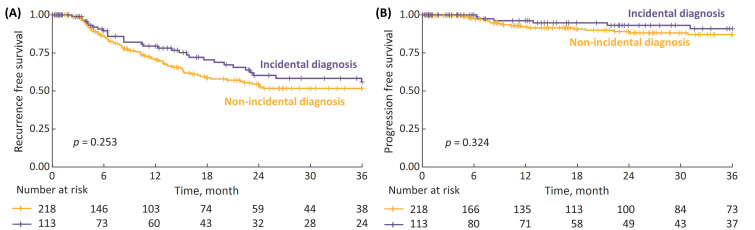
(**A**) 3-year recurrence-free survival (RFS) in non-muscle-invasive bladder cancer (NMIBC) patients stratified by type of diagnosis (incidental vs. non-incidental); and (**B**) 3-year progression-free survival (PFS).

**Table 1 cancers-15-00668-t001:** Patient characteristics.

		Non-Incidental Diagnosis (*n* = 310)	Incidental Diagnosis (*n* = 125)	*p*-Value
Median age, year (IQR)		68 (62–75)	68 (61–75)	0.753
Sex, male		237 (76%)	85 (68%)	0.069
Primary grade	LG	117 (38%)	77 (62%)	<0.001
	HG	193 (62%)	48 (38%)	
Primary stage	NMIBC	218 (70%)	113 (90%)	<0.001
	PUNLMP or Ta	132 (43%)	83 (66%)	0.020
	T1 or CIS ^a^	86 (28%)	30 (24%)	
	MIBC	61 (20%)	7 (6%)	<0.001
	Tx, non-metastatic	21 (7%)	4 (3%)	0.147
	Metastatic	10 (3%)	1 (1%)	0.190
Primary stage (HG only) ^b^	NMIBC	104 (54%)	36 (75%)	0.008
	Ta	27 (9%)	9 (7%)	0.909
	T1 or CIS ^a^	77 (25%)	27 (22%)	
	MIBC	61 (32%)	7 (15%)	0.019
	Tx, non-metastatic	18 (9%)	4 (8%)	0.831
	Metastatic	10 (5%)	1 (2%)	0.698

^a^ Both isolated or concurrent CIS. ^b^ Percentages and *p* values calculated for HG patients only. IQR, interquartile ranges; LG, low-grade; HG, high-grade; NMIBC, non-muscle-invasive bladder cancer; PUNLMP, papillary urothelial neoplasm of low malignant potential; CIS, carcinoma in situ; and MIBC, muscle-invasive bladder cancer.

**Table 2 cancers-15-00668-t002:** Multivariable stepwise Cox regression model investigating associations between specific variables and overall mortality during follow-up.

Variable	HR, 95% CI	*p*-Value
Age, years	1.04, 1.02–1.06	<0.001
Sex (male)	0.98, 0.64–1.52	0.943
HG cancer	2.77, 1.70–4.53	<0.001
MIBC	3.28, 2.18–4.94	<0.001
Metastatic disease ^a^	1.55, 0.69–3.46	0.288
Incidental diagnosis	0.81, 0.52–1.27	0.361

^a^ Either nodal or visceral metastases; HG, high-grade; MIBC, muscle-invasive bladder cancer; HR, hazard ratio; and CI, confidence interval

## Data Availability

The data are available from the corresponding author on request.

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
