# Peer review of "Incidental Diagnosis of Urothelial Bladder Cancer: Associations with Overall Survival"

_cancers, 2023, doi:10.3390/cancers15030668_

Round 1

Reviewer 1 Report

The authors performed an interesting and thorough revision from a retrospective cohort data of consecutive patients with primary diagnosis of urothelial carcinoma, who underwent transurethral resection of bladder tumor (TURBT), and stratified patients based on the criteria that the diagnosis was incidental or non-incidental. The aim of the study was to assess whether incidental diagnosis of bladder tumor was associated with improved overall survival or relative survival in comparison with non-incidental ones. I read the manuscript with interest. According to their results, The Authors concluded that incidental diagnosis of bladder cancer may be associated with higher overall survival (and relative survival) mostly when tumor was detected with ultrasound imaging. Their results also suggest a better survival in incidental muscle-invasive carcinoma subgroup compared with non-incidental counterpart.

The article is overall well written, the subject is actual and interesting for the readers. Although the topic is of interest since literature lacks similar research, there are some aspects to comment:

1) The definition of criteria for incidental diagnosis of bladder cancer should be clarified better. Even if macro or micro haematuria does not necessarily refer specifically to an oncological pathology of the bladder, it is experience of all urologist that a good number of bladder cancer patients may came to attention of urologist with haematuria as first manifestation of disease. Increased awareness of the substantially stronger association of gross hematuria with cancer is a key educational message for clinicians. The authors should clarify better this concept and justify their decision to consider haematuria among the inclusion criteria.

2) Regarding the grading of tumors elucidated in Table 1, it is intriguing to note that the number of HG incidental cases, although lower than the non-incidental group, are high (38%). Given that HG tumors are more commonly associated with positive cytology and macro haematuria, the authors should consider and eventually re-consider the selection criteria for “incidental diagnosis” and perhaps this aspect should be explained in the materials and methods and in the discussion, supported by literature.

3) The authors stated that natural history of bladder carcinoma may consist of an asymptomatic phase, representing an earlier stage of the disease: how were patients stratified in respect of the different signs and symptoms found (no symptoms vs haematuria vs irritative manifestation, etc)? This topic could be summarized in a table.

4) Data of pTa and pT1 in NMIBC should be added in Table 1.

4) Recurrence data of LG tumor patients (incidental vs non-incidental) during the 3 years FU should be shown as well.

5) What kind of treatment has been performed in non-muscle invasive HG carcinoma patients after diagnosis and surgery? Was there a difference in the treatment choice between incidental and non-incidental cases? It is possible that the survival data in HGs was somehow influenced since many of these patients already had another tumor, and not all patients received the same treatment. The authors should make a comment. What about radical treatment for muscle-invasive carcinoma patients?

6) Minor: all cases (more than 700) were diagnosed by a pathologist dedicated to urogenital pathology as reported by the authors, in line with the ISUP classification of tumors. Have been the slides of tumors reviewed before analysis of data? Furthermore, based on the affiliations, it is not clear if the pathologist is a co-author of the manuscript. If no, he might at least deserve the acknowledgments, in consideration of the huge number of cases he analyzed.

Author Response

Dear Reviewer,

We would like to express our sincere gratitude for the time and effort you spent on reviewing our manuscript. Thank you very much for the valuable remarks included in your report.

Below is our point-to-point response to your comments.

Ad. 1. Thank you for raising our attention at this topic. It is worth mentioning that it was our institutional experience of high proportion of incidental, asymptomatic BC patients which made us design and conduct this study. You wrote that “a good number of BC patients may come to an attention of urologist with hematuria as first manifestation of the disease”. This is true and this may be considered a rule. However, we verified and demonstrated that a large number of our patients were in fact diagnosed incidentally. Our proportion only slightly outnumbers the rates showed in other studies. In other reports published by researchers from Polish institutions, 18-22% of BC patients were incidentally diagnosed or presented without hematuria. Also, in a large British study by Price et al., almost 30% of BC patients did not report hematuria. Incidental or asymptomatic diagnosis of BC may be in fact a relatively common phenomenon and we aimed to measure it. Also, please note that our definition was not: “a patient had no symptoms” but “diagnosis was made at an occasion other than evaluating symptoms”. This was aimed to best represent a possible screening-detected BC scenario, in which a symptomatic patient may have not undergone diagnostic evaluation for various reasons, but this incidental diagnosis was their earliest opportunity for establishing diagnosis (however, those were only very single cases in our study group). In line with the presented definition, we considered a patient in whom a diagnosis of a tumor was made in the process of evaluating hematuria a non-incidental case. A patient diagnosed with a bladder tumor, for example, during abdominal ultrasound performed due to intestinal problems, who later admitted having experienced hematuria but had not reported it to a doctor and had not been evaluated for this hematuria, was still considered incidental (however, as mentioned above, those were extremely rare cases). We discussed this issue more broadly in the revised version of the manuscript (Lines: 312-322).

Ad. 2. Thank you for this comment. Yes, HG cancers are more commonly associated with a positive cytology, however, an asymptomatic case diagnosed due positive cytology could not be considered an incidental diagnosis, as performing urine cytology typically is aimed to evaluate a patient already suspected of BC, making this always a non-incidental case. An association of HG (vs LG) with hematuria may seem obvious, but we could not find any literature data supporting this hypothesis. As mentioned in our manuscript, this is our second paper in regard to the incidental diagnosis of BC. In our first paper, aimed to describe the features of incidental BC in more basic detail, we demonstrated that among NMIBC patients, HG (vs LG) was independently negatively associated with incidental diagnosis (OR 0.52, p = 0.009, multivariable analysis). So, in fact, we were the first to report an association of HG with non-incidental diagnosis (mainly, hematuria). We wanted to avoid unnecessary self-citations or duplication of results.

Ad. 3. This is very important and thank you for this comment. As mentioned above, we wanted to avoid duplication of results. Our previous paper was the one aimed to describe the features of incidental and non-incidental BC cases in more detail. However, recognizing that this data is significant, we added it in the revised version of the manuscript, in the Results section and in a supplementary table (Lines: 135-136, Supplementary Table 2).

Ad. 4. Thank you for this comment. We revised the Table 1 accordingly. We included PUNLMP into the Ta subgroup and CIS (isolated or concurrent) to T1 group to facilitate statistical comparison (Chi-square).

Ad. 4. You raised our attention at a very significant issue and thank you for that. Previously we had mentioned that we lack data in regard to recurrence or progression events. However, for the purpose of this revision, we managed to review our database and we did collect necessary data. In the revised version of the manuscript we included 3-year RFS, as well as PFS. We revised other parts of the manuscript accordingly. (Lines: 28-30, 38, 98-105, 113-114, 190-192, 288-292, Figure 6).

Ad. 5. This is a very interesting and important comment. Thank you very much. We have not thought about it this way. I just checked the database. In both cohorts (incidental and non-incidental) the BCG-therapy rate for HG cancer was 38% (almost similar). The number may seem low, but please not that many patients have been referred to other institutions after diagnosis and those 38% represent patients who were given BCG therapy in our center. Due to the high risk of bias, as discussed above, we did not include this data nor analysis in the revised version of the manuscript. Same applies to cystectomies. However, we discussed the issue of possible treatment differences in the Discussion section (Lines: 300-303).

Ad. 6. Thank you for this remark. The slides were not reviewed, we based on the available pathology reports. The cohort consists of patients diagnosed in the course of eight years. It has not been a single pathologist, but rather a team of pathologists. As we are the major cancer center in the country, we work with specialized pathologists, dedicated to organ systems. We included the pathology team in the Acknowledgements.

We have included the above-discussed changes in our revised paper. Also, changes made in line with the other reviewer opinion were made, which primarily consist of a multivariable Cox survival analysis. All the changes are marked with the MS Word “track changes”. We hope that the revised manuscript successfully fulfills your valuable queries and expectations. Once again, thank you for the comprehensive review.

Reviewer 2 Report

The paper is well written, but some methodological concerns exist and results must be interpretted very carefully.

First of all, the authors should provide information about patient's comorbidities and its association with overall survival. Better OS for incidental diagnosis might come from the different prevalence of comorbidities in incidental and symptomatic BC patients. This is a major confounder. Moreover, symptomatic patients can bear a higher burden of bladder cancer (e.g. multiple tumours, large tumours). Information about tumour characteristics should also be added.

Secondly, proportional Cox hazards analyses should be performed to check if there is a true association between incidental diagnosis and survival. Multivariate analysis is necessary to adjust for confounders (e.g. age, comorbidities, tumour size) and check if the incidental diagnosis is really an independent prognostic factor for OS.

Thirdly, other disease-specifc survival (e.g. RFS or PFS or CSS) outcomes must be studied. OS is highly dependent on age and comorbidities in NMIBC patients. 

How was the survival data obtained?

In summary, the paper is well-written but the relevance and authenticity of findings is questionable.

Author Response

Dear Reviewer,

We would like to express our sincere gratitude for the time and effort you spent on reviewing our manuscript. Thank you very much for your valuable comments.

*Please note: The line numbers apply to the document file with "Simple markup" selected at "Track changes" option in MS Word.

You suggested that we should provide information about patient comorbidities and include this factor in our survival analyses. Thank you for this very important comment. Sadly, we lack reliable data in regard to comorbidities and the retrospective design of the study makes it impossible to have this data collected post-hoc. We have thoroughly discussed this limitation in our manuscript (Lines: 293-307). We are aware that this might have significantly impacted the relevance of our findings. However, while this is not meant to serve as evidence, please note that the marked mortality in HG and MIBC, as compared to LG or NMIBC, suggests that the role of non-BC-specific mortality might have been minor. In order to partially overcome this limitation, we included relative survival analyses, which are aimed to diminish the effect of age and sex-related mortality, attributable to specific population. Unfortunately, in regard to the issue you raised, there is nothing more we can change in our study design at this point.

You have also suggested that data in regard to tumor burden (size and number) might have influenced survival of our patients. Thank you for this important remark. Unfortunately, while the TURBT reports are available and we did try to obtain this data, we ultimately decided not to do so, as this would be associated with extreme risk of bias. Most of the reports were written with no uniform protocol, many of them being laconic, and when we realized how unreliable this data was, we decided not to include it. We discussed this limitation in the revised version of the manuscript (Lines: 330-332).

Thank you very much for suggesting that we should have applied proportional Cox hazard analyses. We did so in the revised version of the manuscript. As explained above, we did not include comorbidities and tumor size as possible mortality risk factors, however, we did include other variables: age, sex, disease stage and grade. Please see the updated Results section (Lines: 197-200, Table 2), as well as other related updates (Lines: 30-31, 37-38, 116-118, 238-244, 339-341). As the Kaplan-Meier curves have previously suggested, incidental diagnosis was revealed not to be an independent prognostic factor on multivariable analysis. In fact, as we previously discussed, the improved OS with incidental diagnosis was most probably due to LG cancer overdiagnosis. Please note that at no point we consider our findings to serve as evidence of incidental diagnosis improving survival. Any association was most probably in the mechanism of relative disease downgrading and downstaging, and mortality was grade- and stage-dependent.

You suggested that RFS or PFS or CSS outcomes should be added to our study. Thank you very much for this valuable remark. Previously we had mentioned that we lack data in regard to recurrence or progression events. However, for the purpose of this revision, we managed to review our database and we did collect necessary data. In the revised version of the manuscript, we included 3-year RFS, as well as PFS. We revised other parts of the manuscript accordingly. (Lines: 28-30, 38, 98-105, 113-114, 190-192, 288-292, Figure 6).

Unfortunately, CSS is data we are not able to collect. Patients typically do not die in our institution; they die at home or at a hospice. We are aware of the widespread practice of hospice or general practitioners providing garbage ICD codes for the purpose of the Polish governmental death registry and thus we are very reluctant to proceed with such analyses. You asked how the overall survival data was obtained. Our institutional database is being regularly updated with survival data obtained from governmental census bureau and we used this data in our analyses.

We are aware of the limitations of our study. We do state that the results should be interpreted with great caution. However, still we would like our paper to be considered important, pilot evidence in regard to incidental BC diagnosis. We believe that the changes we have made in the revised version of the manuscript are in line with your expectations. Once again, thank you for the effort of conducting the comprehensive review.

Round 2

Reviewer 1 Report

None.